# Influence of Competition on Anxiety and Heart Rate Variability in Young Tennis Players

**DOI:** 10.3390/healthcare10112237

**Published:** 2022-11-09

**Authors:** Sergio García-Gonzálvez, Daniel López-Plaza, Oriol Abellán-Aynés

**Affiliations:** 1Bachelor Program in Psychology, Faculty of Medicine, UCAM Universidad Católica de Murcia, 30130 Murcia, Spain; 2International Chair of Sport Medicine, UCAM Universidad Católica de Murcia, 30130 Murcia, Spain; 3Faculty of Sport, UCAM Universidad Católica de Murcia, 30130 Murcia, Spain

**Keywords:** anxiety, heart rate variability, athlete, competition, training

## Abstract

The aim of this study was to analyze the effect of competition on anxiety and heart rate variability (HRV) in tennis players. Thirty tennis players (15 boys and 15 girls) were included in the study. In boys, the mean age was 14.53 years, mean height was 169.20 cm and mean weight was 60.60 kg. In girls, the mean age was 13.60 years, mean height was 164.07 cm and mean weight was 53.33 kg. Competitive anxiety was measured using the Competitive Anxiety Inventory-2 (CSAI-2) and the State Trait Anxiety Inventory (STAI). HRV was also analyzed using a validated HR chest band (Polar H7). These measurements were performed at two different times, before training and before competing, both times maintaining the same conditions for the participants. In addition, a gender differentiation was carried out. Somatic anxiety and state anxiety were significantly higher at the pre-competition time compared to the pre-training time (*p* < 0.05). On the contrary, the rest of the variables did not present significant differences between conditions (*p* > 0.05). On the other hand, no correlation was observed between anxiety and heart rate variability either before competition or before training. As for gender differentiation, significant differences were obtained between males and females in state anxiety and heart rate, identifying higher values in females.

## 1. Introduction

Nowadays, anxiety disorders show higher prevalence than in the last decades of the past century [1]. Anxiety, according to the American Psychiatric Association [2] in the Diagnostic and Statistical Manual of Mental Disorders, Fifth Edition (DSM-V), is defined as a state of excessive worry or apprehensive anticipation that is related to different moments or events. Some of its most common symptoms are fatigue, muscle tension, sleep problems or palpitations [3]. Professionals indicate that the number of people suffering from anxiety has increased, since in a study conducted, it was reported that treatment in these disorders increased by almost 75% [4]. In a meta-analysis conducted by Racine et al. (2021) [5] at a global level, it was observed that the prevalence of anxiety symptoms in adolescents stands at 20.5%, being higher in females.

The analysis of the prevalence of anxiety in sport can be considered a difficult task due to inequalities in definitions, severity and assessment tools applied [6]. The most basic definition of competitive anxiety is given by Mellalieu et al. (2020) [7], which defines it as the anxiety associated with a sport situation or competition. Anxiety during competition is a negative emotional reaction such as worry, tension or fear, that arises in an athlete who sees how their self-esteem is threatened by a situation during competitions that they cannot control or defeat [8]. Anxiety pertaining to the athlete’s personality would be known as “trait anxiety”, and when it pertains to thoughts or feelings outside the athlete, it would be “state anxiety” [8].

Sport anxiety in tennis players is related to the maturational evolution of each athlete, both in personal and performance-related factors. In tennis, anxiety increases as the athlete grows [9]. This might be partially explained by the fact that between the ages of 12 and 20, young people find themselves in adolescence, a transition period characterized by numerous changes in their behaviors and ways of thinking [9]. In this same study, it was found that 50% of the young athletes suffered from considerable anxiety, with the percentage and severity increasing with age [9]. In another study conducted in badminton, it was observed that competitive anxiety in adolescents and the fear of having anxiety symptomatology negatively influences the performance of the athlete [10]. The importance of psychological factors in this type of sport has been previously reported [11], since 80% of the victories of professional athletes are determined by this aspect, but highlighting competitive anxiety [8,11].

Changes in the physiological activity of athletes can be reflected in the heart rate (HR), known to be one of the most widely used non-invasive methods in the assessment and analysis of cardiac activity. At rest, the heartbeat in healthy individuals behaves in an unstable or variable manner, which is commonly referred to as heart rate variability (HRV). HRV is defined as the variation in heartbeat frequency over a defined time interval [12]. The relationship between these two parameters is inversely proportional with respect to intensity and workload, i.e., the higher the HR, the more HRV decreases. HRV shows the result of the interactions of the autonomic nervous system (ANS) and the cardiovascular system, which allows an analysis of ANS activity in a non-invasive, painless, simple and inexpensive way for researchers [12,13,14].

Bisschoff et al. (2016) [15] highlighted the importance of taking into account mood-related variables when HRV is measured in competitive periods, as this can help to understand how anxiety reacts in competitive situations. In tennis, this is of great importance because it is known that its practice improves cardiac sympathetic–vagal balance, which causes a reduction in the SNS (somatic nervous system) and, in turn, the development of cardiovascular diseases in the general population [16]. Ref. [17] observed that the emotional indexes and cortisol in tennis players were higher in competition than in training, observing, likewise, a reduction in HRV in competition. Although a correlation between anxiety and psychophysiological states revealed no significant differences, it is recommended to monitor HRV to improve performance before competition and training [17].

Previous studies conducted in other sports identified a relationship between competitive anxiety and HRV [13,18,19]. Furthermore, the measurement of HRV provides an individual, practical and complementary tool to measure competitive anxiety that is useful for coaches to optimize sport performance [19]. The main objective of this study was to analyze the differences in competitive anxiety and HRV between competition and training in male and female tennis players.

## 2. Materials and Methods

### 2.1. Participants

A total of 30 adolescent tennis players (14.07 ± 2.00 years of age), 15 males (14.53 ± 2.42 years of age) and 15 females (13.60 ± 1.40 years of age), voluntarily participated in the present study with a prospective ex post facto design. All participants were experienced players (>3 years of experience) and trained on a regular basis (>6 h of training per week).

None of the athletes had any physical injuries nor were they taking any medication at the time of the measurements. In addition, none of the participants had any reason that prevented them from participating in the study. We used a non-probabilistic method of convenience to select the sample from clubs in the same geographical area to obtain as homogeneous a sample as possible in terms of performance and level of competition. The parents or legal guardians of the participants signed a consent form in which the objective and method of the study were explained in detail. In addition, participants were informed of the possibility of withdrawing from the study at any time and without giving any reason. Previously, the current investigation had been approved by the Ethics Committee of the Catholic University of San Antonio de Murcia with the code CE092204.

### 2.2. Variables

Variables such as age, height and weight were measured. These were described separately for males and females. The mean age of men was 14.53 ± 2.42 years of age, mean height was 169.20 ± 14.05 cm and mean weight was 60.60 ± 17.20 kg, whereas females presented a mean age of 13.6 ± 1.40 years of age, mean height of 164.07 ± 5.10 cm and mean weight of 54.33 ± 7.69 kg. 

### 2.3. Instruments

#### 2.3.1. H7 Polar

All measurements of R-wave time intervals (RR) were performed using a validated [20] Polar H7 (Kempele, Finland) heart rate sensor. This monitor recorded variations in interbeat intervals. The Polar heart rate chest strap was adjusted according to the size of each participant’s chest, just below the lower line of the pectoralis major muscle and after the electrodes were moistened with water to facilitate electrical transmission. RR time series were processed through a smartphone via the Elite HRV mobile application [21] to export RR interval data. Subsequent analyses of RR intervals were performed using Kubios HRV version 3.0 software [22]. Using this software, artifacts presented in the time series were identified throughout very low, low or medium threshold filters for subsequent removal. With the use of Kubios, we calculated the mean RR intervals, the standard deviation of consecutive RR intervals (SDNN), the root mean square of successive differences between RR intervals (RMSSD), and the percentage of consecutive RR intervals that differ by more than 50 ms (pNN50). The SDNN gives us the long-term variability of RR intervals while RMSSD and pNN50 give us information about the short-term variations in RR intervals. On the other hand, we obtained the stress score (SS) calculated between the relationship between RMSSD ($\mathrm{RMSSD}/\sqrt{2}$) and the inverse value of the SDNN *(*$\frac{1}{\mathrm{SDNN}})$. With the value of SS interpreted as the sympathetic activity, we calculated the sympathetic/parasympathetic ratio (S/PS) using $\mathrm{RMSSD}/\sqrt{2}$ as the parasympathetic value. For the RR intervals, we proceeded with fast Fourier transformation to switch data from a time domain to a frequency domain. From the frequency domain, we extracted the total power for the high-frequency power (HF) for the values between 0.15 Hz and 0.4 Hz, represented based on its natural logarithm (HF_ln_). We repeated the process for frequencies between 0.04 Hz and 0.15 Hz to obtain the low-frequency power (LF) in order to calculate the ratio LF/HF.

#### 2.3.2. Competitive Anxiety

CSAI-2 was used to measure competitive anxiety. This inventory was developed by Martens et al. (1990) [23] to measure the intensity of cognitive and somatic responses and self-confidence before training and before competition. The CSAI-2 consists of 27 items, with 9 items in each of the 3 subscales (cognitive anxiety, somatic anxiety and self-confidence). Participants were asked to rate the intensity of each symptom on a Likert scale ranging from 1 (not at all) to 4 (very much), resulting in scores ranging from 9 to 36 for each subscale [23]. In the subscales of somatic and cognitive anxiety, exceeding 18 points could begin to be considered a problem; in the case of self-confidence, not reaching this score would be a problem [23]. For the analysis, each of the scores was transformed into percentages, so that a higher percentage indicates a higher level of anxiety [23]. This instrument shows homogeneity, reliability and sensitivity in the measurement of psychological characteristics in athletes [23]. Satisfactory internal consistency for the intensity of the subscales has been previously reported with Cronbach’s alpha coefficients ranging from 0.79 to 0.90 [24].

#### 2.3.3. Trait Anxiety and State Anxiety

To measure trait anxiety and state anxiety, we used the Spanish adaptation of the “Cuestionario de ansiedad estado-rasgo” (State-Trait Anxiety Iventory, STAI; [25]). The STAI is a self-evaluation questionnaire composed of 40 items measuring two subscales of anxiety: trait anxiety (20 items) and state anxiety (20 items). Each of these was completed by the participants using a 4-point Likert response system of intensity from 0 (not at all) to 3 (very much so), which resulted in a total score between 0 and 60 points for each of the subscales. In the samples of the Spanish population, internal consistency levels ranging between 0.84 and 0.93 Cronbach’s alpha coefficients were found. This questionnaire is one of the most widely used both nationally and worldwide. This is also reflected in the fact that although it was first adapted to Spanish more than three decades ago, this questionnaire keeps being used both for research and in clinical practice.

### 2.4. Procedures

During March and April 2022, before one of their training sessions, heart rate variability was measured with a Polar H7 device for 5 min in the supine position, and at complete rest in a soundproofed room at a temperature of 24 °C and a relative humidity of 52%. Once the HRV measurement was completed, questionnaires were completed by participants. An identical process was carried out again 3 days later before a competition, taking the measurements at the same time of day and in the same conditions as before training. The time elapsed between measurements before training and before competing was the same for all participants, always at the same time of the day and immediately before the game. For this reason, we always took both measurements the week of the competition. In addition, this competition was of great relevance for the athletes since; apart from playing for themselves, they also played for the team due to this being a team competition.

### 2.5. Statistical Analysis

Statistical analysis was performed using the statistical analysis software package for social sciences version 24.0 (SPSS, Inc., Chicago, IL, USA). The normality of the distribution of the variables was analyzed using the Shapiro–Wilk test while the descriptive statistics were calculated by extracting the mean and standard deviation of each variable. A two-factor analysis of variance (ANOVA) for repeated measurements was used to analyze the effect of competition on anxiety and HRV between group genders (male and female) and during two conditions (pre-training and pre-competition). A Bonferroni post hoc test was used when significant interactions were observed (group × condition). Partial eta squared (η^2^_p_) was calculated as the effect size. An η^2^_p_ between 0.1 and 0.24 was considered a low effect; between 0.25 and 0.36 was considered a medium effect; and >0.37 was considered a high effect. The significance limit was set at *p* < 0.05. This analysis was performed with the aim to compare between groups such as gender, and conditions such as pre-competition and pre-training.

Subsequently, a linear correlation analysis between variables was performed. Depending on the distribution of the sample, Pearson’s (r) or Spearman’s (rho) correlation index were used. A correlation value of <0.2 was interpreted as no correlation; 0.2–0.4 was interpreted as low correlation; 0.4–0.6 was interpreted as medium correlation; 0.6–0.8 was interpreted as high correlation; and >0.8 was interpreted as very high correlation. Correlations were interpreted in the same way in the case of negative correlations. We carried out this analysis in order to observe the interdependence between the different outcomes.

## 3. Results

Table 1 shows the general description of the sample between groups (boys and girls).

The correlation between anxiety variables and HRV variables are presented in Table 2. Pearson’s correlation index indicates low-to-no association between anxiety and HRV variables, whereas high and very high relationships were identified between the different subscales of each variable.

Table 3 shows the effect of competition and gender on anxiety. The values of the mean, standard deviation and *p*-value of the comparison are presented. The values of the mean and *p*-values indicate that state anxiety is significantly higher in both genders during competition than during training, as is somatic anxiety in females. State anxiety is also significantly higher in females than in males. Furthermore, competition significantly influences state anxiety and somatic anxiety.

Table 4 shows the effect of competition and gender on HRV. The values of the mean, standard deviation and *p*-value of the comparison are presented. The values of the mean and *p*-values indicate that heart rate is significantly higher in females than in males. In males, RMSSD is significantly lower during competition compared to training. Furthermore, competition significantly influences HR.

## 4. Discussion

This study provides normative data on competitive anxiety and heart rate variability in two important moments of tennis players’ routines (pre-training and pre-competition). The main hypothesis of a relationship between HRV and competitive anxiety was rejected, whereas the hypotheses of competitive anxiety being higher during pre-competition and HRV being higher during pre-training are discussed below.

Most of the previous investigations concerning competition and anxiety have used the CSAI-2 to measure competitive anxiety [17,18,26,27,28]. However, in the present study, STAI was also used in order to corroborate the validity of the data provided by the CSAI-2. From this questionnaire, the state anxiety scale was analyzed and taken into consideration for the final results. Since all the subscales measured by the CSAI-2 assess anxiety at the same moment (state anxiety), the results provided by trait anxiety would not be significant. Similar to the findings from [17], no correlation between competitive anxiety and HRV was identified. This could be due to the fact that the anxiety perceived by the athletes was not the same as that reflected in the HRV data.

Regarding the different parameters of HRV, anxiety is related to an increase in LF and LF/HF and, conversely, with a decrease in HF [13]. It should be emphasized that an impairment in HRV may require a longer rest time in order to return and compete at a high level since competitive performance is related to HRV [18]. Thus, although competitive anxiety might indeed lead to impaired HRV after competing, performance might not be affected [18].

LF/HF and somatic anxiety were found to be related to a low/medium level of anxiety [17], confirming that pre-training also promotes psychophysiological changes. Even though no statistical correlation was detected, the influence of physiological changes on psychological states can be established in the current investigation. Therefore, it is recommended to monitor HRV to increase performance in both pre-training and pre-competition [17].

Regarding anxiety variables, [18] reported no relationship between HRV and self-confidence but a significant and positive association of cognitive anxiety with somatic anxiety. Conversely, the findings here are far from being able to establish a relationship between any of the HRV variables and cognitive and somatic anxiety. A continuous exposure to competitive stress allows habituation with the specific environment of competition, which possibly also reduces the flight response and the perception of subjective anxiety, in turn decreasing cognitive and somatic anxiety and increasing self-confidence [19].

According to Mateo et al. (2012) [19], anxiety is the only variable that changes significantly. During pre-training, anxiety remains low and stable until it reaches its peak just before the start of the competition. In another similar study conducted in swimmers, identical results were obtained, observing significant differences in somatic anxiety but non-significant differences in cognitive anxiety and self-confidence [13]. Other authors have also reported a significant increase in cognitive anxiety [26]. Therefore, although only state anxiety and somatic anxiety are significantly higher, we can confirm the hypothesis that competitive anxiety is superior during pre-competition than during pre-training [13,17,26,29].

Regarding gender differentiation and in agreement with Martínez-Gallego et al. (2022) [30], significantly higher state anxiety values were determined in female tennis players. Similarly, girls reported higher cognitive and somatic anxiety, and less self-confidence than boys according to Filaire et al. (2009) [31] and the results presented here.

Only a few studies were found concerning HRV before training and before competition. Regardless of the sport, most HRV and related performance investigations have been conducted and focused on pre- and post-match analyses [15,26,28]. Some of the common aspects of these studies showed a significant increase in HR and a significant decrease in RMSSD when compared pre- and post-competition. A reduced HRV prior to competition could have different effects on the athlete. Meanwhile, the parasympathetic system is reduced while there is an increased activity of the sympathetic system. This may be because the competition is interpreted by the athlete as a stressful situation or a problem to be solved that produces changes in the physiological responses of the organism based on the alert situation. In this way, a reduced HRV before the competition can provoke benefits in the athlete by preparing their physiological systems to respond to the stressful situation in various ways such as a higher heart rate, a greater muscular contractile capacity or a way to react more immediately to physical demands that can be key during the development of the match. A point against presenting reduced HRV values before the competition may cause a more impulsive reaction of the athlete during the competition and a less strategic approach; however, to analyze this fact would require more research that relates, in addition to HRV and anxiety, the athlete’s personality with their way of playing. In the event that HRV may have detrimental effects on performance, recent studies have found that techniques such as neurofeedback training can help increase HRV in people suffering from anxiety and depression in a clinical setting [32,33]. However, in the case of neurofeedback training used in athletes and the increase in HRV having a beneficial effect pre-competition, it would be interesting to analyze how this low HRV and anxiety affect the results in competition.

Prior studies in racquet sports have identified similar HR behavior in competition to that determined here when compared to training [15,28], revealing significant increases in HR before pre-competition.

After the analysis of the results, the previous HRV hypothesis is confirmed since HRV values are lower before competition than before a training session. In addition, LF/HF decreased but not significantly, as also reported by similar studies [17]. This might be due to the fact that, during competition, both psychological and physiological demands increase, which causes an increase in HR and a decrease in HRV [17].

In the study of the different variables of HRV, RMSSD stands out as it is one of the best that reflects HRV, which in our study, decreases, although not significantly. Despite this, other variables such as pNN50, HFIn, SS and S/PS decrease significantly pre-competition compared with pre-training [19,26]. SDNN, another important variable, is reduced significantly, as was found in the study carried out during a mountain biking competition [19]. The cause of this change is due to the pre-competition anxiety that a real competition generates, which decompensates HR and slows vagal control (SDNN) [19].

In the current investigation, several limitations were encountered that prevented us from obtaining all the data without any bias. Almost all of them are related to HRV because this variable is influenced by numerous external factors. Another important limitation of our study is a low sample size due to having such a specific population, with it being difficult to find participants with the same characteristics and practiced in the same competition on the same days with the same conditions.

In future research, it would be of great interest to increase the sample size to provide more validity to our study. Dividing the athletes by levels could lead to more significant scores; therefore, finding professional tennis players to participate could be beneficial to the study. Furthermore, for a future potential line of investigation, an additional post-competition measurement to compare it with pre-competition might be taken into consideration. Similarly, pre- and post-competition examination could be performed and related to final performance. Examining the difference between an individual and a team competition might also be interesting for future research.

## 5. Conclusions

Competition provokes an impairment of state and somatic anxiety in female youth tennis players. This impairment is observed in male youth tennis players in state anxiety but not in the rest of anxieties assessed. Additionally, an impairment in cardiac autonomic control is observed in female youth tennis players with an increase in heart rate before a competition. This effect was not observed in male youth tennis players.

## Figures and Tables

**Table 1 healthcare-10-02237-t001:** Descriptive variables of the sample.

Variables	Boys		Girls		
	M	SD	M	SD	*p*
Age (years)	14.53	2.42	13.6	1.4	0.2
Height (cm)	169.2	14.05	164.07	5.09	0.213
Weight (kg)	60.6	17.2	54.33	7.69	0.209

**Table 2 healthcare-10-02237-t002:** Correlation between anxiety variables and heart rate variability variables.

	STA	COA	SOA	SCONF	HR	SDNN	RMSSD	SS
STA	1	0.271	0.538 *	−0.305	0.003	0.137	0.108	−0.025
COA		1	0.749 *	−0.318	0.038	0.156	0.154	0.164
SOA			1	−0.430 *	0.23	0.115	0.075	0.266
SCONF				1	−0.163	−0.146	−0.106	−0.073
HR					1	−0.588 *	−0.612 *	0.620 *
SDNN						1	0.839 *	−0.790 *
RMSSD							1	−0.556 *
SS								1

STA: State anxiety; COA: Cognitive anxiety; SOA: Somatic anxiety; SCONF: Self-confidence; HR: Heart rate; SDNN: Standard deviation; RMSSD: Root mean square value; SS: Stress index. *: *p* < 0.05.

**Table 3 healthcare-10-02237-t003:** Effects of competition and gender on anxiety.

							ANOVA (F. p. ŋ^2^_p_)								
		Training		Competition			Competition Effect			Gender Effect			Competition × Gender Effect		
Outcome	Gender	M	SD	M	SD	*p*	F	*p*	ŋ^2^_p_	F	*p*	ŋ^2^_p_	F	*p*	ŋ^2^_p_
STA	Female	11.87	8.77	23.60 *	9.86	<0.001	22.713	<0.001	0.448	4.949	0.034	0.15	1.399	0.247	0.048
	Male	8.87	8	15.93	7.15	0.017									
COA	Female	20.33	7.2	22.73	5.96	0.059	3.312	0.079	0.106	1.919	0.177	0.064	0.937	0.341	0.032
	Male	18.2	6	18.93	6.05	0.552									
SOA	Female	15.8	5.24	18.33	6.02	0.019	5.806	0.023	0.172	2.31	0.14	0.076	1.237	0.276	0.042
	Male	14.27	3.59	15.2	3.1	0.367									
SCONF	Female	27.6	6.47	25.33	5.86	0.066	2.932	0.098	0.095	1.62	0.213	0.055	0.991	0.328	0.034
	Male	29.2	4.53	28.6	5.78	0.616									

STA: State anxiety; COA: Cognitive anxiety; SOA: Somatic anxiety; SCONF: Self-confidence. * *p* < 0.05 compared to male.

**Table 4 healthcare-10-02237-t004:** Effects of competition and gender on heart rate variability.

							ANOVA (F. p. ŋ^2^_p_)								
		Training		Competition			Competition Effect			Gender Effect			Competition × Gender Effect		
Outcome	Gender	M	SD	M	SD	*p*	F	*p*	ŋ^2^_p_	F	*p*	ŋ^2^_p_	F	*p*	ŋ^2^_p_
HR	Female	80.00	12.70	87.93 *	13.42	0.007	6.594	0.016	0.191	2.601	0.118	0.085	2.527	0.123	0.083
	Male	75.93	16.26	77.80	9.21	0.495									
SDNN	Female	68.12	32.92	58.02	20.50	0.131	0.739	0.397	0.026	0.03	0.863	0.001	1.797	0.191	0.060
	Male	60.94	27.80	68.68	29.62	0.736									
RMSSD	Female	70.11	45.83	43.32	29.13	0.005	3.478	0.073	0.111	0.025	0.874	0.001	5.931	0.021	0.175
	Male	52.65	33.01	54.37	38.39	0.69									
pNN50	Female	32.35	25.95	18.73	19.84	0.140	1.418	0.244	0.048	0.129	0.723	0.005	0.916	0.347	0.032
	Male	27.63	22.38	25.48	21.82	0.870									
HF_ln_	Female	6.77	1.52	6.26	1.43	0.095	1.451	0.238	0.049	1.088	0.306	0.037	1.532	0.226	0.052
	Male	530.90	2030.85	6.62	1.23	0.982									
LF/HF	Female	1.39	1.49	2.93	2.65	0.146	2.889	0.1	0.094	0.015	0.902	0.001	0.173	0.681	0.006
	Male	524.53	2024.97	2.58	3.60	0.372									
SS	Female	14.93	6.89	15.43	8.08	0.938	0.424	0.520	0.015	0.714	0.405	0.025	0.291	0.594	0.1
	Male	15.32	6.39	13.13	5.26	0.407									
S/PS	Female	0.91	1.81	1.18	2.32	0.753	0.193	0.664	0.007	0.411	0.527	0.014	0.788	0.382	0.027
	Male	0.99	1.54	0.51	0.44	0.356									

HR: Heart rate; SDNN: Standard deviation; RMSSD: Root mean square value; pNN50: percentage of consecutive RR intervals that differ more than 50 ms; HF_ln_: High-frequency power based on its natural logarithm; LF/HF: Ratio of low-frequency/high frequency power; SS: Stress score; S/PS: Sympathetic/parasympathetic ratio. * *p* < 0.05 compared to male.

## Data Availability

All data presented in the current study are available upon request.

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
