# Peer review of "Influence of Competition on Anxiety and Heart Rate Variability in Young Tennis Players"

_healthcare, 2022, doi:10.3390/healthcare10112237_

Round 1
Reviewer 1 Report
Major revisions must be made to this article for it to merit publication in a journal of this impact:
-Why have young tennis players been chosen as a specific population? Please justify
-What type of study is it? When was it done?
-The sample size is too small (n=30). Why not more population? How was the sample size calculated?
- It shows that the study was approved by an ethics committee, but there is no evidence? add the code.
- Provide more information on the platforms and tests used to measure the variables.
- The statistics being used, and the underlying purpose for the various analyses is confusing and needs to be made clear
-Discussion section. I recommend adding a section with the limitations of the study. Also, Could you suggest any future lines to expand the internal and external validity of this research?
-As a result of some of the previous comments, specially the scope of this paper, the conclusions do not seem to be very relevant. What does this article contribute to scientific knowledge? What are the news? Why are the results of this article useful?
In the discussion or the conclusions section, these questions should be answered.
-Check text formatting and references. For example, there is an error between lines 62-63.
Author Response
Reviewer #1
Major revisions must be made to this article for it to merit publication in a journal of this impact:
-Why have young tennis players been chosen as a specific population? Please justify
Thank you for your suggestion, since it was not entirely clear why we chose this population. We chose adolescent tennis players as a specific population mainly because we observed that in the studies previously reviewed, this period of their life was marked by changes that could be affecting their performance. The fact of being in tennis players is due to the lack of research on these athletes and the influence of psychological factors in this sport.
We added the following sentence in this section:
In this same study it was found that 50% of the young athletes suffered from considerable anxiety, with the percentage and severity increasing with age [9].
-What type of study is it? When was it done?
We had not been aware that we had not added what type of study it was, so I would like to thank you for your observation. We have added both questions in the method section:
A total of 30 adolescent tennis players (14.07 ± 2.00 years of age), 15 males 14.53 ± 2.42 years of age) and 15 females (13.60 ± 1.40 years of age) voluntarily participated in the present study with a prospective ex-post facto design
And:
During March and April 2022 (line 159).
-The sample size is too small (n=30). Why not more population? How was the sample size calculated?
We conducted a convenience sample collection in a number of clubs in the same geographical area. In this way, we were able to analyze athletes of similar performance and participating in competitions of similar level to make the sample more homogeneous and obtain more practical results. In the geographical area where the study was conducted, there were no other players of this age and at this level of competition. For this reason, we used a non-probabilistic method to obtain the sample for convenience and did not calculate the sample size based on the total population.
We added the following sentence in the methods section:
We used a non-probabilistic method of convenience to select the sample from clubs in the same geographical area in order to obtain a sample as homogeneous as possible in terms of performance and level of competition.
- It shows that the study was approved by an ethics committee, but there is no evidence? add the code.
Thank you for noticing the lack of information, we have added the code of the ethics committee
with the code CE092204
- Provide more information on the platforms and tests used to measure the variables.
We have added one more sentence for each test used to make it clearer why it is being used. Also, we have detailed more deeply the measurement and analysis of heart rate variability.
In the sub-scales of somatic and cognitive anxiety, exceeding 18 points could begin to be considered a problem; in the case of self-confidence, not reaching this score would be a problem [23].
This questionnaire is one of the most widely used both worldwide and nationally. This is also reflected in the fact that although it was first adapted to Spanish more than 3 decades ago, it continues to be used both for research and in clinical practice.
Thanks for your suggestion, these improve the quality of the work.
- The statistics being used, and the underlying purpose for the various analyses is confusing and needs to be made clear
We have added these two sentences after each of the statistical descriptions to make it more clear which aim of the research is being answered with each of the analyses
This analysis was performed with the aim to compare between groups such as gender and conditions such as pre-competition and pre-training.
We carried out this analysis in order to observe the interdependence between the different outcomes.
-Discussion section. I recommend adding a section with the limitations of the study. Also, Could you suggest any future lines to expand the internal and external validity of this research?
We have expanded the section on limitations and future research, thank you for your advice to add more.
Another important limitation of our study is a low sample size because of having such a specific population, being difficult to find participants with the same characteristics and practiced in the same competition on the same days with the same conditions.
In future research it would be of great interest to increase the sample size to pro-vide more validity to our study. Dividing the athletes by levels could lead to more sig-nificant scores, so finding professional tennis players to participate could be beneficial to the study.
Examining the difference between an individual and a team competition might also be interesting for future research.
-As a result of some of the previous comments, specially the scope of this paper, the conclusions do not seem to be very relevant. What does this article contribute to scientific knowledge? What are the news? Why are the results of this article useful?
As you have said, the conclusion section was not quite clear, so I have deleted the following sentences:
On the other hand, no correlation was found between HRV and anxiety either before training or before competition. Despite this and due to the significance of somatic anxiety, coaches are advised to work on anxiety and HR control during training to improve performance during both training and competition.
And added new ones:
Competition provokes an impairment of state and somatic anxiety on female youth tennis players. This impairment is observed in male youth tennis players in state anxiety but not in the rest of anxieties assessed. Also, an impairment of cardiac auto-nomic control is observed in female youth tennis players with an increase in heart rate before a competition. This effect was not observed in male youth tennis players.
In the discussion or the conclusions section, these questions should be answered.
-Check text formatting and references. For example, there is an error between lines 62-63.
Text formatting and references have been checked. Thank you for your valuables comments.

Reviewer 2 Report
Dear authors,
Congratulations on your work! Hopefully, I have some comments and suggestions to help improve your paper.
I first had to mention the references citation in the body text. Please, do not start a sentence with the reference number. For example: "[15] highlighted the importance of taking into account (...)" - Line 63 - or "Anxiety according to [2] in the Diagnostic and Statistical Manual (...)" - Line 27. It should be "Bisschoff and colleagues (2016)[15] highlighted the importance of taking into account (...)" and "Anxiety according to the American Psychiatric Association (2013) [2] in the Diagnostic and Statistical Manual (...)".
In the methodology:
- Had all athletes a competition 3 days after the first measure? How important was that competition (it certainly influenced the HRV outcome)? Were the HRV measures taken immediately before the match, or with what time/interval difference?
- Explain the RMSS variable (you can even place the equation here).
- you should have a section only for Statistical Analysis.
Results:
- Table 1. can have another column with the p-value between groups.
- Table 3. must be in a landscape view.
Discussion:
It can be essential to discuss the impact of different training programs. I'm assuming (since it is not implicit in the methodology) that the athletes have different coaches, so they have various training programs.
So, you found out that the HRV is lower before a pre-competition. How much can that influence the athlete's performance?
I know that your work is not about what to do with lower levels of HRV, but for readers would be enjoyable to know what can be done to contrary that. From a psychology point of view, I'm afraid I cannot help, however, it is already known that neurofeedback training can be an important tool helping to enhance HRV RMSSD (1Domingos et al., 2022; 2White et al., 2017). It is only a suggestion, but I would add one or two psychological "tools" or training to help improve HRV and mention alternative techniques to do it, too (such as the neurofeedback training discussed above).
1Domingos, C., Silva, C. M. D., Antunes, A., Prazeres, P., Esteves, I., & Rosa, A. C. (2021). The influence of an alpha band neurofeedback training in heart rate variability in athletes. International Journal of Environmental Research and Public Health, 18(23), 12579.
https://doi.org/10.3390/ijerph182312579
2White, E. K., Groeneveld, K. M., Tittle, R. K., Bolhuis, N. A., Martin, R. E., Royer, T. G., & Fotuhi, M. (2017). Combined neurofeedback and heart rate variability training for individuals with symptoms of anxiety and depression: A retrospective study. NeuroRegulation, 4(1), 37-37.
https://doi.org/10.15540/nr.4.1.37
Best of luck!
Author Response
Reviewer #2
Dear authors,
Congratulations on your work! Hopefully, I have some comments and suggestions to help improve your paper.
I first had to mention the references citation in the body text. Please, do not start a sentence with the reference number. For example: "[15] highlighted the importance of taking into account (...)" - Line 63 - or "Anxiety according to [2] in the Diagnostic and Statistical Manual (...)" - Line 27. It should be "Bisschoff and colleagues (2016)[15] highlighted the importance of taking into account (...)" and "Anxiety according to the American Psychiatric Association (2013) [2] in the Diagnostic and Statistical Manual (...)".
Thanks. All the format and references has been reviewed
In the methodology:
- Had all athletes a competition 3 days after the first measure? How important was that competition (it certainly influenced the HRV outcome)? Were the HRV measures taken immediately before the match, or with what time/interval difference?
At all times we have maintained the same conditions for all participants. Measurements were always taken in the same week for training and competition with the same days of difference. Measurements took place immediately before the game so that there were as little bias as possible. Thanks to your appreciation, we are aware of this shortcoming in the study and have been able to add the following information:
The time elapsed between measurements before training and before competing was the same for all participants, always at the same time of the day and immediately before the game. For this reason, we always took both measurements the week of the competi-tion. In addition, this competition was of great relevance for the athletes since, apart from playing for themselves, they also played for the team because of being a team competition.
- Explain the RMSS variable (you can even place the equation here).
Thank you for your comment. We have added a description of HRV variables for making it easier for readers to interpret the data. We added the following paragraph on the methods section. We didn’t add the formula on the paper but we added the explanation of how it is calculated and its interpretation. However, if you think it is important to add the formula in the manuscript we can do it.
With the use of Kubios we calculated the mean RR intervals, the standard deviation of consecutive RR intervals (SDNN), the root mean square of successive differences between RR intervals (RMSSD) and the percentage of consecutive RR intervals that differ more than 50ms (pNN50). The SDNN gives us the long-term variability of RR intervals while RMSSD and pNN50 gives us information about the short-term variations of RR intervals. On the other hand we obtained the stress score (SS) calculated between the relationship between RMSSD () and the inverse value of the SDNN (. With the value of SS interpreted as the sympathetic activity, we calculated the sympathetic/parasympathetic ratio (S/PS) using as the parasympathetic value. For the RR intervals, we proceeded with fast Fourier transformation to switch data from a time-domain to frequency-domain. From the frequency domain we extracted the total power for the high frequency power (HF) for the values between 0.15 Hz and 0.4 Hz and represented based on its natural logarithm (HFln). We repeated the process for frequencies between 0.04 Hz and 0.15 Hz to obtain the low frequency power (LF) in order to calculate the ratio LF/HF-
you should have a section only for Statistical Analysis.
Thank you for noticing this issue. We made a mistake uploading the final draft of the paper and we removed the heading of the section involuntarily. We added the heading
Results:
- Table 1. can have another column with the p-value between groups.
We have added the p value comparing descriptive values between gender groups
|
Table 1. Descriptive variables of the sample. |
|||||
|
Variables |
Boys |
Girls |
|
||
|
M |
SD |
M |
SD |
p |
|
|
Age (years) |
14.53 |
2.42 |
13.60 |
1.40 |
0.2 |
|
Height (cm) |
169.20 |
14.05 |
164.07 |
5.09 |
0.213 |
|
Weight (kg) |
60.60 |
17.20 |
54.33 |
7.69 |
0.209 |
- Table 3. must be in a landscape view.
We changed the view for that table and now it is in landscape format. Now the table is easier to read and to interpret. We did the same with table 4. Thanks for noticing
Discussion:
It can be essential to discuss the impact of different training programs. I'm assuming (since it is not implicit in the methodology) that the athletes have different coaches, so they have various training programs.
We totally agreed with that. All participants belonged to two local clubs with similar level and they all performed similar training volume per week. We added the following to clarify that
All participants were experienced players (>3 years of experience) and trained on regular basis (>6 hours of training per week).
So, you found out that the HRV is lower before a pre-competition. How much can that influence the athlete's performance?
Thank you for your appreciation. We have added the following paragraph on the discussion section taking also into account your comments about the possibility of introducing the concept of neurofeedback training.
A reduced HRV prior to competition could have different effects on the athlete. On the one hand, the parasympathetic system is reduced while there is an increased activity of the sympathetic system. This may be because the competition is interpreted by the athlete as a stressful situation or a problem to be solved that produces changes in the physiological responses of the organism based on the alert situation. In this way, a reduced HRV before the competition can provoke benefits in the athlete by preparing his physiological systems to give responses to the stressful situation such as a higher heart rate or others such as a greater muscular contractile capacity or a way to react more immediately to physical de-mands that can be key during the development of the match. A point against presenting reduced HRV values before the competition may cause a more impulsive reaction of the athlete during the competition and a less strategic approach, however, to analyze this fact would require more research that relates, in addition to HRV and anxiety, the athlete's personality with the way of playing. In the event that HRV may have detrimental effects on performance, recent studies have found that techniques such as neurofeedback training can help increase HRV in people suffering from anxiety and depression in a clinical set-ting [32,33]. However, in case neurofeedback training is used in athletes and the increase in HRV has a beneficial effect pre-competition, it would be interesting to analyze how this low HRV and anxiety affects the results in competition.
I know that your work is not about what to do with lower levels of HRV, but for readers would be enjoyable to know what can be done to contrary that. From a psychology point of view, I'm afraid I cannot help, however, it is already known that neurofeedback training can be an important tool helping to enhance HRV RMSSD (1Domingos et al., 2022; 2White et al., 2017). It is only a suggestion, but I would add one or two psychological "tools" or training to help improve HRV and mention alternative techniques to do it, too (such as the neurofeedback training discussed above).
1Domingos, C., Silva, C. M. D., Antunes, A., Prazeres, P., Esteves, I., & Rosa, A. C. (2021). The influence of an alpha band neurofeedback training in heart rate variability in athletes. International Journal of Environmental Research and Public Health, 18(23), 12579.
https://doi.org/10.3390/ijerph182312579
2White, E. K., Groeneveld, K. M., Tittle, R. K., Bolhuis, N. A., Martin, R. E., Royer, T. G., & Fotuhi, M. (2017). Combined neurofeedback and heart rate variability training for individuals with symptoms of anxiety and depression: A retrospective study. NeuroRegulation, 4(1), 37-37.
https://doi.org/10.15540/nr.4.1.37
Best of luck!

Round 2
Reviewer 1 Report
accept